# Unsat Core Prediction through Polarity-Aware Representation Learning over Clause-Literal Hypergraphs

Zhenchao Sun [1]   Shuai Ma [1]   Ping Lu [1]   Chongyang Tao [1]

## Abstract

Graph neural networks have been widely used in Boolean satisfiability (SAT) tasks to learn structural information from SAT formulas. The goal of these studies is to solve SAT instances or to enhance SAT solvers, including tasks such as unsat-core prediction. However, most existing approaches model a SAT formula as a bipartite graph or a directed acyclic graph, which are less direct in capturing clause-level and higher-order interactions among literals and clauses. Moreover, these approaches are limited in modeling intrinsic polarity-related properties of SAT, such as the complementary relationship between the positive and negative literals of a variable. To address these limitations, we propose a polarity-aware representation learning framework over clause-literal hypergraphs. We model SAT formulas as clause-literal hypergraphs augmented with a clause incidence graph to capture higher-order structural interactions. We then introduce a polarity-aware decomposition mechanism that separates variable representations into polarity invariant and equivariant components, explicitly modeling the relationship between positive and negative literals, with the resulting literal representations propagated along the hypergraph structure. We further incorporate a polarity-inversion consistency regularization to reinforce polarity-consistent representations during training. Experimental results on multiple SAT datasets demonstrate the effectiveness of the proposed approach.

## 1. Introduction

The Boolean satisfiability problem (SAT) is a foundational problem in computer science that impacts many research fields, such as planning (Büttner & Rintanen, 2005), verification (Vizel et al., 2015), and security (Mironov & Zhang, 2006). This is the first problem that was proven to be NP-complete (Cook, 1971). Over the past decades, significant progress has been made in SAT solving, particularly with the development of conflict-driven clause learning (CDCL) solvers. CDCL solvers are complete and capable of solving large, complex real-world instances (Biere et al., 2021). However, such effectiveness heavily depends on heuristics that rely on human domain expertise.

Recently, many studies have explored learning-based approaches to solve SAT instances or enhance SAT solving (Bünz & Lamm, 2017; Selsam et al., 2019; Amizadeh et al., 2019; Li et al., 2023; Cameron et al., 2020; Selsam & Bjørner, 2019; Shi et al., 2023; Wang et al., 2024). Graph neural networks (GNNs) have demonstrated impressive capability in learning from structured data (Kipf & Welling, 2017; Hamilton et al., 2017), making them well suited for representing SAT formulas and capturing their structural properties. A representative line of work leverages GNNs to guide the heuristics of CDCL solvers by learning structural information from SAT formulas (Selsam & Bjørner, 2019; Shi et al., 2023; Wang et al., 2024). For instance, some studies model SAT formulas as bipartite graphs and predict unsat-core variables, i.e., variables involved in unsatisfiable cores, to guide variable selection in CDCL solvers (Selsam & Bjørner, 2019; Shi et al., 2023). Others predict backbone variables to initialize the phase selection heuristic (Wang et al., 2024), while additional work applies learning-based models to guide other solver heuristics, such as restarts and clause deletion (Liang et al., 2018; Liu et al., 2024).

Although existing studies have demonstrated effectiveness, these learning-based approaches have several limitations. One such limitation is that existing GNN-based SAT models focus on representation learning from graph structures, and overlook intrinsic properties of the SAT formulation. In SAT problems, each variable is associated with a pair of complementary literals with inverse polarity, corresponding to its positive and negative occurrences. However, existing approaches typically treat literals and clauses as independent graph nodes and form variable representations by directly combining positive and negative literal embeddings,

[1]SKLCCSE Lab, Beihang University, Beijing, China. Correspondence to: Shuai Ma <mashuai@buaa.edu.cn>.

*Proceedings of the 43rd International Conference on Machine Learning*, Seoul, South Korea. PMLR 306, 2026. Copyright 2026 by the author(s).

which weakens the modeling of polarity inversion between them (Zhang et al., 2020; Selsam & Bjørner, 2019). Some methods introduce additional edges between complementary literals (Selsam et al., 2019), while others model variables and clauses as nodes, and encode literal occurrences and their polarities via edge labels (Kurin et al., 2020; Li & Si, 2022; Wang et al., 2024). Although these designs enable message passing between literals, they do not explicitly enforce two key constraints: (i) complementary literals of the same variable should share common information, and (ii) their representations should reflect the polarity inversion relationship. How to effectively incorporate these constraints into model design and training remains an open challenge for learning SAT-related representations.

Beyond literal-level constraints, another limitation is that SAT formulas are often modeled as bipartite graphs or directed acyclic graphs, which limits their capacity to capture higher-order literal and clause interactions. On the one hand, each clause contains multiple literals, and assignments to one literal can influence other literals within the same clause, leading to multi-literal dependencies that cannot be fully captured by pairwise connections. On the other hand, clauses also interact with each other, and satisfiability often depends on dependencies among multiple clauses rather than clause pairs, as reflected in unsat cores (Shi et al., 2023). In bipartite graph representations, message passing is restricted to local neighborhoods, making higher-order and long-range dependency modeling rely on deep architectures that may suffer from over-smoothing. Hypergraphs provide a more natural representation for such structures by directly modeling multi-literal clause constraints via hyperedges, and have been adopted in prior work on SAT-related problems (Feng et al., 2019; Chen et al., 2025).

To address these limitations and model higher-order dependencies in SAT formulas, we propose a **P**olarity-**A**ware representation learning framework over clause–literal hypergraphs for un**SAT** core prediction (**PASAT**). The framework represents SAT formulas as hypergraphs augmented with a clause incidence graph, and employs a polarity-aware decomposition message passing mechanism that separates variable representations into invariant and equivariant components, capturing the relationship between positive and negative literals of the same variable. To further enforce polarity-consistent representations, we introduce a polarity-inversion consistency regularization, which leverages polarity-flipped formulas as an alternative view during training.

Our main contributions are summarized as follows:

- We propose a polarity-aware representation learning framework over hypergraphs for SAT formulas, augmented with a clause incidence graph, to capture higher-order relationships among literals and clauses.

- We introduce a polarity-aware decomposition mechanism that separates invariant and equivariant components of variable representations, enabling effective modeling of the inherent relationship between positive and negative literals of the same variable.

- We develop a polarity-aware consistency regularization, which leverages polarity-flipped formulas as an alternative view to enhance learning for SAT problems.

- Extensive experiments on multiple SAT datasets demonstrate that the proposed approach achieves improved performance on unsat core variable prediction compared with existing baselines.

## 2. Preliminaries

A Boolean formula $\phi$ is composed of Boolean variables combined by logical operators. A Boolean variable $v_i$ can take only two values: *true* or *false* $(1/0)$, and the logical operators include conjunction $(\wedge)$, disjunction $(\vee)$, and negation $(\neg)$. A Boolean formula can be converted into conjunctive normal form (CNF), which is a conjunction of clauses, where each clause is a disjunction of literals (Biere et al., 2021). Each literal $l$ is either a Boolean variable $v_i$ or its negation $\neg v_i$, referred to as the positive and negative literals of $v_i$, respectively. The two literals have opposite polarity. We denote the number of Boolean variables as $N$ and the number of clauses as $M$, and use $\mathcal{L} = \{l_0, \ldots, l_{2N-1}\}$ to denote the set of all literals and $\mathcal{C} = \{c_0, \ldots, c_{M-1}\}$ to denote the set of clauses. The goal of the SAT problem is to determine whether there exists an assignment of the variables such that the Boolean formula $\phi$ evaluates to *true*. If this is the case, the formula is called *satisfiable*; otherwise, it is *unsatisfiable*. A variable is called an unsat-core variable if at least one of its literals appears in an unsatisfiable core.

## 3. Methodology

In this section, we present our polarity-aware representation learning framework over hypergraphs for SAT formulas. The framework consists of a hypergraph representation with clause-level structure, a decomposed message passing mechanism for variable representations, and a polarity-inversion consistency regularization for training. An overview of the proposed framework is illustrated in Figure 1.

### 3.1. Hypergraph-Based Representation for CNF

We propose a hypergraph-based SAT formula modeling method that represents a CNF formula as a hypergraph and incorporates an additional clause incidence graph (CIG) to facilitate message passing among clauses.

**Hypergraph Representation for CNF Formulas.** A hypergraph is a generalization of a graph in which hyperedges

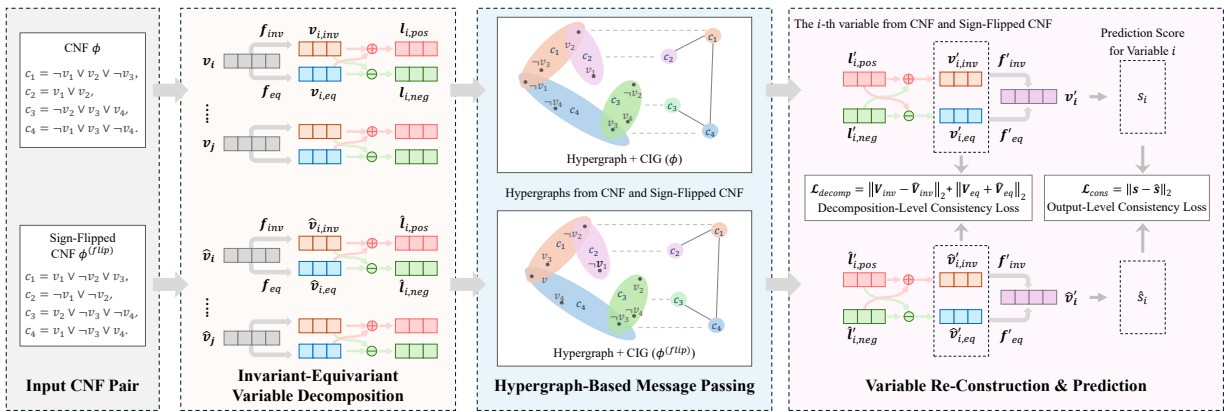

*Figure 1.* Overview of the proposed polarity-aware hypergraph-based framework (PASAT). Given a CNF formula $\phi$, the representation of a variable $\mathbf{v}_i$ is decomposed into a polarity-invariant component $\mathbf{v}_{i,\text{inv}}$ and a polarity-equivariant component $\mathbf{v}_{i,\text{eq}}$. These components are combined to construct positive and negative literal embeddings $\mathbf{l}_{i,\text{pos}}$ and $\mathbf{l}_{i,\text{neg}}$. The literal embeddings are then propagated on the hypergraph to obtain updated embeddings $\mathbf{l}'_{i,\text{pos}}$ and $\mathbf{l}'_{i,\text{neg}}$. In parallel, a polarity-flipped formula $\phi^{(\text{flip})}$ is processed using shared parameters. The resulting representations from the two views are reconstructed into variable embeddings $\mathbf{v}'_i$ and $\hat{\mathbf{v}}'_i$, which are used to produce prediction scores $s_i$ and $\hat{s}_i$, with training guided by task supervision and polarity-inversion consistency regularization.

are allowed to connect more than two nodes (Feng et al., 2019). To capture higher-order relationships between literals and clauses, we represent a CNF formula $\phi$ as a hypergraph $\mathcal{H} = (\mathcal{V}_H, \mathcal{E}_H)$, where $\mathcal{V}_H$ denotes the node set and each node $u_i$ corresponds to a literal $l_i$. The hyperedge set $\mathcal{E}_H$ represents clauses in the formula, with each hyperedge $e_j$ corresponding to a clause $c_j$. An incidence relation is defined such that a node corresponding to literal $l_i$ is connected to hyperedge $e_j$ if $l_i$ appears in clause $c_j$. This clause–literal relationship is encoded by an incidence matrix $\mathbf{H} \in \mathbb{R}^{2N \times M}$, where $N$ is the number of variables and $M$ is the number of clauses. Specifically, $\mathbf{H}_{ij} = 1$ if literal $l_i$ appears in clause $c_j$, and 0 otherwise.

In order to model relationships among different clauses, we further extend our representation by constructing a clause incidence graph (CIG), denoted as $\mathcal{G}_C = (\mathcal{V}_C, \mathcal{E}_C)$. Each node $u_j^C \in \mathcal{V}_C$ corresponds to a hyperedge $e_j \in \mathcal{E}_H$, and thus represents the clause $c_j$ in the CNF formula $\phi$. Two nodes in $\mathcal{V}_C$ are connected by an edge in $\mathcal{G}_C$ if and only if the corresponding clauses share at least one common literal. Formally, we define the edge weight as $w_{ij}^C = |\mathcal{L}(c_i) \cap \mathcal{L}(c_j)| \,/\, |\mathcal{L}(c_i) \cup \mathcal{L}(c_j)|$, where $\mathcal{L}(c_i)$ denotes the set of literals appearing in clause $c_i$.

**Hypergraph-Based Message Passing.** Based on $\mathcal{H}$ and $\mathcal{G}_C$, we design a message passing mechanism that propagates information between literals and clauses while explicitly modeling clause–clause interactions.

Let $\mathbf{L}^{(t)} \in \mathbb{R}^{2N \times d}$ denote the literal representations at message passing round $t$. Literal–clause message propagation is performed via the incidence matrix $\mathbf{H} \in \mathbb{R}^{2N \times M}$ following the hypergraph convolution formulation in (Bai et al., 2021). Specifically, information is first aggregated from literals to clauses and then propagated back to literals through a degree-normalized hypergraph operator:

$$\mathbf{M}_{\mathrm{H}}^{(t)} = \mathbf{D}^{-1}\mathbf{H}\mathbf{B}^{-1}\mathbf{H}^{\top}\mathbf{L}^{(t)}\mathbf{W}^{(t)}, \tag{1}$$

where $\mathbf{W}^{(t)} \in \mathbb{R}^{d \times d}$ is a learnable weight matrix. $\mathbf{D} \in \mathbb{R}^{2N \times 2N}$ and $\mathbf{B} \in \mathbb{R}^{M \times M}$ denote the literal and clause degree matrices, defined as

$$\mathbf{D}_{ii} = \sum_{j=1}^{M} \mathbf{H}_{ij}, \qquad \mathbf{B}_{jj} = \sum_{i=1}^{2N} \mathbf{H}_{ij}. \tag{2}$$

However, this formulation captures clause interactions only implicitly through literals, which may limit its ability to directly characterize higher-order clause dependencies.

To address this limitation, we extend message passing to the clause incidence graph $\mathcal{G}_C$. Let $\mathbf{A}_C \in \mathbb{R}^{M \times M}$ denote its weighted adjacency matrix and $\mathbf{D}_C$ its degree matrix. Clause-level message propagation is defined as

$$\begin{aligned} \mathbf{C}^{(t)} &= \mathbf{B}^{-1}\mathbf{H}^{\top}\mathbf{L}^{(t)}\mathbf{W}^{(t)}, \\ \Delta\mathbf{C}^{(t)} &= \mathbf{D}_C^{-1/2}\mathbf{A}_C\mathbf{D}_C^{-1/2}\mathbf{C}^{(t)}\mathbf{U}, \end{aligned} \tag{3}$$

where $\mathbf{U} \in \mathbb{R}^{d \times d}$ is a learnable weight matrix. The refined clause representations are obtained by

$$\mathbf{C}'^{(t)} = \mathbf{C}^{(t)} + \alpha\,\sigma\!\left(\Delta\mathbf{C}^{(t)}\right), \tag{4}$$

with a learnable scaling parameter $\alpha$ and a non-linear activation function $\sigma(\cdot)$. The refined clause information is then propagated back to literals by

$$\mathbf{M}^{(t)} = \mathbf{D}^{-1}\mathbf{H}\mathbf{C}'^{(t)}. \tag{5}$$

Finally, each literal representation is updated by combining its current embedding, the aggregated clause message, and the representation of its negated literal:

$$\mathbf{L}^{(t+1)} = f_{\text{update}}(\mathbf{L}^{(t)}, \mathbf{M}^{(t)}, \bar{\mathbf{L}}^{(t)}), \tag{6}$$

where $\bar{\mathbf{L}}^{(t)}$ is a row-wise permutation of $\mathbf{L}^{(t)}$ that provides the embeddings of the corresponding complementary literals. This update function enables the model to jointly capture clause-induced dependencies and polarity-level constraints. By performing multiple rounds of such message passing, the model incrementally integrates higher-order clause structure into the literal representations.

### 3.2. Invariant–Equivariant Variable Decomposition

While the hypergraph-based message passing mechanism introduced above captures structural dependencies between literals and clauses and generates literal-level representations, we still need to obtain variable-level representations from literal representations. Existing studies usually directly concatenate the positive and negative literal representations and feed them into MLPs to produce variable representations. However, these methods neglect intrinsic properties at the variable level in SAT problems. For example, each pair of complementary literals has opposite polarity. Such polarity relations are independent of clauses and cannot be fully characterized by literal–clause message propagation alone. Moreover, a pair of complementary literals should share certain common information, reflecting the fact that they originate from the same Boolean variable.

To explicitly model this property, we introduce a polarity-aware decomposed message passing mechanism for Boolean variables. This mechanism operates at the variable level and is tightly coupled with the literal-level message passing described in the previous subsection.

Let $\mathbf{V}^{(t)} \in \mathbb{R}^{N \times 2d}$ denote the matrix of variable representations at round $t$, where each row $\mathbf{v}_i^{(t)} \in \mathbb{R}^{2d}$ corresponds to a Boolean variable $v_i$. At initialization, variable representations can be randomly initialized, or set to $\mathbf{v}_i^{(0)} = \mathbf{1}_{2d}$ for all $i \in \{0, \dots, N-1\}$, yielding the initial matrix $\mathbf{V}^{(0)} \in \mathbb{R}^{N \times 2d}$. Each variable representation is decomposed into two $d$-dimensional components:

$$\mathbf{V}_{\text{inv}}^{(t)} = f_{\text{inv}}^{(t)}(\mathbf{V}^{(t)}), \qquad \mathbf{V}_{\text{eq}}^{(t)} = f_{\text{eq}}^{(t)}(\mathbf{V}^{(t)}), \quad (7)$$

where $\mathbf{V}_{\text{inv}}^{(t)}, \mathbf{V}_{\text{eq}}^{(t)} \in \mathbb{R}^{N \times d}$, and $f_{\text{inv}}^{(t)}(\cdot)$ and $f_{\text{eq}}^{(t)}(\cdot)$ are learnable mappings. For variable $v_i$, its invariant and equivariant components are denoted as $\mathbf{v}_{i,\text{inv}}^{(t)}$ and $\mathbf{v}_{i,\text{eq}}^{(t)}$, respectively. The invariant component encodes polarity-invariant information shared by both literals and captures intrinsic properties of the variable, such as membership in the unsat core. The equivariant component captures polarity-sensitive information that changes sign under Boolean negation.

Based on this polarity-aware decomposition, the representations of the positive and negative literals associated with variable $v_i$ are constructed as

$$\mathbf{l}_{v_i}^{(t)} = \mathbf{v}_{i,\text{inv}}^{(t)} + \mathbf{v}_{i,\text{eq}}^{(t)}, \qquad \mathbf{l}_{\neg v_i}^{(t)} = \mathbf{v}_{i,\text{inv}}^{(t)} - \mathbf{v}_{i,\text{eq}}^{(t)}. \quad (8)$$

The literal representation matrix $\mathbf{L}^{(t)} \in \mathbb{R}^{2N \times d}$ can be obtained by stacking all literal vectors. Under this ordering, the positive and negative literals of variable $v_i$ correspond to rows $2i$ and $2i + 1$ in $\mathbf{L}^{(t)}$, respectively. That is,

$$\mathbf{L}_{2i}^{(t)} = \mathbf{l}_{v_i}^{(t)}, \qquad \mathbf{L}_{2i+1}^{(t)} = \mathbf{l}_{\neg v_i}^{(t)}. \quad (9)$$

This literal representation matrix serves as the input to the hypergraph-based message passing. After message propagation over the hypergraph and the clause incidence graph, the updated literal representations are denoted as $\mathbf{L}^{(t+1)}$. The variable representations are then recovered by recombining the updated literal representations. For each variable $v_i$, the invariant and equivariant components are recovered as

$$\begin{aligned} \mathbf{v}_{i,\text{inv}}^{(t+1)} &= \tfrac{1}{2}\left(\mathbf{L}_{2i}^{(t+1)} + \mathbf{L}_{2i+1}^{(t+1)}\right), \\ \mathbf{v}_{i,\text{eq}}^{(t+1)} &= \tfrac{1}{2}\left(\mathbf{L}_{2i}^{(t+1)} - \mathbf{L}_{2i+1}^{(t+1)}\right). \end{aligned} \quad (10)$$

Finally, the variable representation matrix is updated by transforming and concatenating the recovered components:

$$\mathbf{V}^{(t+1)} = \left[ f_{\text{inv}}'^{(t)}(\mathbf{V}_{\text{inv}}^{(t+1)}), \; f_{\text{eq}}'^{(t)}(\mathbf{V}_{\text{eq}}^{(t+1)}) \right], \quad (11)$$

where $f_{\text{inv}}'^{(t)}(\cdot)$ and $f_{\text{eq}}'^{(t)}(\cdot)$ are learnable mappings and $[\cdot, \cdot]$ denotes column-wise concatenation.

### 3.3. Training Objective

We adopt the same training objective as in prior work on NeuroCore (Selsam & Bjørner, 2019). Specifically, the model is trained to predict a score for each variable, indicating its likelihood of appearing in an unsatisfiable core.

The prediction scores for all variables are produced by applying a linear projection $g(\cdot)$ to the corresponding invariant representations after the final round of message passing (after $T$ rounds)

$$\mathbf{s} = g\left(f_{\text{inv}}'(\mathbf{V}_{\text{inv}}^{(T)})\right), \quad (12)$$

Then $\mathbf{s}$ can be passed to the softmax function to define a probability distribution $\mathbf{p}$ over variables.

For each unsatisfiable training instance, the label is given as a binary indicator of whether a variable appears in an unsatisfiable core. Following (Selsam & Bjørner, 2019), we normalize this indicator to obtain a target distribution $\mathbf{p}^*$ by assigning uniform probability to all variables in the core and zero probability to the others. The training objective minimizes the Kullback–Leibler divergence (Kullback & Leibler, 1951) between the target distribution and the predicted distribution,

$$\mathcal{L}_{\text{core}} = D_{\text{KL}}(\mathbf{p}^* \| \mathbf{p}) = \sum_{i=0}^{N-1} p_i^* \log \frac{p_i^*}{p_i}. \quad (13)$$

This loss encourages the model to assign higher scores to variables that are more likely to participate in an unsatisfiable core, and is optimized jointly with all learnable components of the network.

### 3.4. Polarity-Inversion Consistency Regularization

In previous sections, we focused on representation learning for SAT formulas and variable-level prediction tasks. We now turn to a training-level constraint motivated by intrinsic properties of SAT problems.

**Properties of Literal Polarity Inversion.** We first study two intrinsic properties of SAT formulas. Polarity inversion refers to flipping the polarity (sign) of all literals in a CNF formula while preserving the variable set and clause structure. Given a CNF formula $\phi = (x_1 \vee x_2) \wedge (\neg x_3 \vee \neg x_4)$, we invert the polarity of all literals while keeping all logical operations unchanged, obtaining a sign-flipped CNF formula $\phi^{(\text{flip})} = (\neg x_1 \vee \neg x_2) \wedge (x_3 \vee x_4)$. At the variable level, polarity inversion exhibits the following two properties.

*Property* 1. Given a CNF formula $\phi$ and its sign-flipped counterpart $\phi^{(\text{flip})}$, the satisfiability of the formula remains unchanged. Moreover, variable-level properties that depend on the structural information of the formula are invariant under polarity inversion. For example, if a set of variables constitutes an unsat-core variable set in an unsatisfiable formula $\phi_{\text{unsat}}$, the same variables also constitute an unsat-core variable set in $\phi_{\text{unsat}}^{(\text{flip})}$.

*Property* 2. For properties related to variable assignments, polarity inversion preserves variable identities while flipping their assignments. This property is also exploited in NeuroBack (Wang et al., 2024), which constructs dual formulas by negating all backbone variables in the original formula. The labels of the resulting sign-flipped formulas correspond to the negated phases of the backbone variables.

**Polarity-Inversion Consistency for Training.** Motivated by the above properties, we introduce a polarity-inversion-based consistency training strategy to encourage the model to learn intrinsic structural properties of SAT problems. For each CNF formula $\phi$ in the training set, we construct a sign-flipped counterpart $\phi^{(\text{flip})}$ as an alternative view. Both $\phi$ and $\phi^{(\text{flip})}$ are processed by the same network with shared parameters, producing variable-level predictions for each view, and the standard variable-level prediction loss is applied independently to both views.

To encourage polarity-aware representations, we further introduce a consistency regularization loss between predictions from the two views. This loss enforces polarity-invariant information (*Property 1*) to remain unchanged and polarity-equivariant information (*Property 2*) to transform consistently under polarity inversion. For the unsat-core prediction task, the prediction vectors $\mathbf{s}$ and $\mathbf{s}^{(\text{flip})}$ from $\phi$ and

$\phi^{(\text{flip})}$ are encouraged to be similar, since polarity inversion does not change unsat-core variables. We denote the corresponding prediction vectors as $\mathbf{s}$ and $\mathbf{s}^{(\text{flip})}$, respectively. The consistency loss is defined as

$$\mathcal{L}_{\text{cons}} = \frac{1}{N} \left\| \mathbf{s} - \mathbf{s}^{(\text{flip})} \right\|_2^2, \tag{14}$$

where $N$ is the number of variables.

In addition to the output-level consistency, we further propose a decomposition-level consistency constraint to explicitly model invariant and equivariant components under polarity inversion, as introduced in Section 3.2.

For each CNF formula, we use the representations from the final message passing round. Under polarity inversion, the invariant components of the original formula $\phi$ and the flipped formula $\phi^{(\text{flip})}$ are expected to remain similar, while the equivariant components are expected to differ. Accordingly, we define the following consistency loss

$$\begin{aligned}
\mathcal{L}_{\text{decomp}} = \frac{1}{N} \sum_{i=0}^{N-1} \Big[ & \left\| \mathbf{v}_{i,\text{inv}}^{(T)} - \mathbf{v}_{i,\text{inv}}^{(T)(\text{flip})} \right\|_2^2 \\
& + \left\| \mathbf{v}_{i,\text{eq}}^{(T)} + \mathbf{v}_{i,\text{eq}}^{(T)(\text{flip})} \right\|_2^2 \Big].
\end{aligned} \tag{15}$$

Combining the above losses, the overall training objective for a single CNF formula is

$$\mathcal{L} = \mathcal{L}_{\text{core}} + \lambda_{\text{cons}} \mathcal{L}_{\text{cons}} + \lambda_{\text{decomp}} \mathcal{L}_{\text{decomp}}, \tag{16}$$

where $\mathcal{L}_{\text{core}}$ denotes the sum of variable-level prediction losses for $\phi$ and $\phi^{(\text{flip})}$, $\mathcal{L}_{\text{cons}}$ and $\mathcal{L}_{\text{decomp}}$ denote the polarity-aware regularization losses, and $\lambda_{\text{cons}}$ and $\lambda_{\text{decomp}}$ weight the two regularization terms.

### 3.5. Combination with SAT Solvers

Finally, we integrate our neural model into CDCL solvers to guide the SAT solving process. Specifically, we follow the NeuroCore (Selsam & Bjørner, 2019) pipeline, where the predicted variable scores are used to periodically overwrite the solver's variable activity scores during solving. While NeuroCore invokes the neural model multiple times during solving to update these scores, our approach runs the neural model only once before solving and passes the predicted scores to the CDCL solver as additional input. During solving, the variable activity scores are periodically adjusted using the same predicted scores, without re-invoking the neural model, thereby reducing GPU computation overhead. In this way, our integration can be viewed as a one-shot scoring strategy that reuses precomputed scores during solving.

## 4. Experiments

In this section, we present an experimental analysis of unsat-core prediction to justify the effectiveness of our method.

*Table 1.* Unsat-core prediction performance on three datasets. Avg. denotes the average performance over available difficulty levels.

| DATASET | METHOD | PRECISION | | | | PR-AUC | | | | ROC-AUC | | | |
|---|---|---|---|---|---|---|---|---|---|---|---|---|---|
| | | EASY | MEDIUM | HARD | AVG. | EASY | MEDIUM | HARD | AVG. | EASY | MEDIUM | HARD | AVG. |
| SR | GCN | 0.845 | 0.857 | 0.938 | 0.880 | 0.891 | 0.894 | 0.961 | 0.915 | 0.703 | 0.715 | 0.778 | 0.732 |
| | NEUROCORE | 0.865 | 0.874 | 0.945 | 0.895 | 0.917 | 0.922 | 0.974 | 0.938 | 0.769 | 0.789 | 0.856 | 0.805 |
| | SATFORMER | 0.847 | 0.859 | 0.938 | 0.881 | 0.895 | 0.900 | 0.964 | 0.920 | 0.709 | 0.726 | 0.787 | 0.741 |
| | PASAT (OURS) | 0.896 | 0.900 | 0.956 | 0.917 | 0.950 | 0.953 | 0.985 | 0.963 | 0.839 | 0.863 | 0.913 | 0.872 |
| CA | GCN | 0.221 | 0.147 | 0.144 | 0.171 | 0.305 | 0.182 | 0.158 | 0.215 | 0.546 | 0.505 | 0.502 | 0.518 |
| | NEUROCORE | 0.335 | 0.191 | 0.189 | 0.238 | 0.443 | 0.272 | 0.285 | 0.333 | 0.672 | 0.573 | 0.571 | 0.605 |
| | SATFORMER | 0.292 | 0.136 | 0.126 | 0.185 | 0.404 | 0.186 | 0.150 | 0.246 | 0.616 | 0.488 | 0.477 | 0.527 |
| | PASAT (OURS) | 0.424 | 0.307 | 0.493 | 0.408 | 0.553 | 0.406 | 0.575 | 0.511 | 0.759 | 0.666 | 0.784 | 0.736 |
| PS | GCN | 0.828 | 0.710 | 0.681 | 0.740 | 0.898 | 0.766 | 0.717 | 0.794 | 0.787 | 0.774 | 0.776 | 0.779 |
| | NEUROCORE | 0.852 | 0.749 | 0.721 | 0.774 | 0.920 | 0.809 | 0.766 | 0.832 | 0.826 | 0.826 | 0.835 | 0.829 |
| | SATFORMER | 0.840 | 0.731 | 0.696 | 0.756 | 0.910 | 0.788 | 0.735 | 0.811 | 0.808 | 0.801 | 0.806 | 0.805 |
| | PASAT (OURS) | 0.864 | 0.775 | 0.752 | 0.797 | 0.930 | 0.838 | 0.803 | 0.857 | 0.845 | 0.861 | 0.880 | 0.862 |

### 4.1. Experimental Settings

**Datasets.** We use three generated SAT datasets, SR, Community Attachment (CA), and Popularity-Similarity (PS), constructed following G4SATBench (Li et al., 2024). Each dataset is divided into easy, medium, and hard levels. For each level, the training set contains 80,000 SAT and UNSAT instance pairs, while the validation and test sets each contain 10,000 pairs. For unsat-core prediction, we use only UNSAT instances for both training and evaluation, following NeuroCore (Selsam & Bjørner, 2019). Detailed dataset descriptions are provided in Appendix A.

**Baselines and Metrics.** We follow the unsat-core prediction setting in G4SATBench (Li et al., 2024) and compare our method with representative baselines, including GCN (Kipf & Welling, 2017), NeuroCore (Selsam & Bjørner, 2019), and SATFormer (Shi et al., 2023). Although many studies have explored learning-based methods for guiding SAT solvers (Kurin et al., 2020; Liu et al., 2024; Wang et al., 2024; Tönshoff & Grohe, 2025), these approaches adopt different pipelines and solver configurations. As a result, direct comparisons with these methods are not feasible, since they do not provide comparable intermediate predictions such as unsat-core variables. Models are evaluated using Top-$M$ Precision, PR-AUC, and ROC-AUC, where $M$ denotes the number of unsat-core variables in each instance.

**Implementation Details.** In our experiments, the number of message-passing rounds on the hypergraph is set to 4. The regularization term $\lambda_{cons}$ is selected from $[0.1, 0.3]$, and $\lambda_{decomp}$ is selected from $[0.05, 0.15]$. All baseline methods follow their released implementations with default parameter settings. The full implementation details and hyperparameter configurations are provided in Appendix B.1.

### 4.2. Experimental Results

We next present our findings.

**Exp-1: Comparison with Existing Methods for Unsat-Core Prediction.** We first compare the unsat-core predic-

tion performance of our method with that of the baseline models. The results are reported in Table 1.

Across all datasets, our method consistently achieves the best unsat-core variable prediction performance compared with the baseline models on all difficulty splits. On the SR, CA, and PS datasets, our method improves the average Precision by approximately 2.5%, 71.4%, and 3.0% over NeuroCore, respectively. Notably, the CA dataset, on which our method achieves the largest performance gains, is particularly challenging for representation learning. Due to its community structure, the associations between variables and clauses exhibit clear clustering patterns, resulting in a more complex topology than those in the SR and PS datasets. In addition, CA instances typically have a larger clause-to-variable ratio and a smaller unsat-core size, which leads to lower overall performance on this dataset. In contrast, PASAT models SAT formulas as a hypergraph augmented with a clause incidence graph, enabling the model to capture such complex structural information. Moreover, the proposed invariant–equivariant variable decomposition and polarity-inversion consistency regularization explicitly model the relationships between complementary literals. As a result, PASAT achieves the best performance on the CA dataset and exhibits the largest performance gains among all datasets. The improvements in PR-AUC further indicate that our method produces more accurate and stable ranking scores for unsat-core variables. Overall, these experimental results justify the design of our method. Additional results on SATCOMP datasets and inference-time/memory usage are provided in Appendix C.1 and C.2, respectively.

**Exp-2: Ablation Study.** We conduct an incremental ablation study to evaluate the effectiveness of the components in our framework. Starting from a bipartite-graph baseline based on NeuroCore, we first replace bipartite graph modeling with hypergraph-based message passing, and then incorporate invariant–equivariant variable decomposition and polarity-inversion consistency regularization. Accordingly, we evaluate four configurations with different component combinations: BIPARTITE GRAPH (NEUROCORE), PASAT

*Table 2.* Ablation via incremental integration of hypergraph message passing (HG), invariant–equivariant variable decomposition (DE), and polarity-inversion consistency regularization (REG). Avg. denotes the average performance over available difficulty levels.

| DATASET | METHOD | PRECISION | | | | PR-AUC | | | | ROC-AUC | | | |
|---|---|---|---|---|---|---|---|---|---|---|---|---|---|
| | | EASY | MEDIUM | HARD | AVG. | EASY | MEDIUM | HARD | AVG. | EASY | MEDIUM | HARD | AVG. |
| SR | BIPARTITE GRAPH (NEUROCORE) | 0.865 | 0.874 | 0.945 | 0.895 | 0.917 | 0.922 | 0.974 | 0.938 | 0.769 | 0.789 | 0.856 | 0.805 |
| | PASAT (HG) | 0.867 | 0.879 | 0.947 | 0.898 | 0.919 | 0.927 | 0.975 | 0.940 | 0.768 | 0.798 | 0.858 | 0.808 |
| | PASAT (HG+DE) | 0.885 | 0.894 | 0.954 | 0.911 | 0.940 | 0.946 | 0.984 | 0.956 | 0.812 | 0.844 | 0.906 | 0.854 |
| | PASAT (HG+DE+REG, FULL) | 0.896 | 0.900 | 0.956 | 0.917 | 0.950 | 0.953 | 0.985 | 0.963 | 0.839 | 0.863 | 0.913 | 0.872 |
| CA | BIPARTITE GRAPH (NEUROCORE) | 0.335 | 0.191 | 0.189 | 0.238 | 0.443 | 0.272 | 0.285 | 0.333 | 0.672 | 0.573 | 0.571 | 0.605 |
| | PASAT (HG) | 0.335 | 0.263 | 0.482 | 0.360 | 0.443 | 0.360 | 0.564 | 0.456 | 0.680 | 0.626 | 0.768 | 0.691 |
| | PASAT (HG+DE) | 0.391 | 0.272 | 0.486 | 0.383 | 0.520 | 0.367 | 0.567 | 0.485 | 0.721 | 0.645 | 0.782 | 0.716 |
| | PASAT (HG+DE+REG, FULL) | 0.424 | 0.307 | 0.493 | 0.408 | 0.553 | 0.406 | 0.575 | 0.511 | 0.759 | 0.666 | 0.784 | 0.736 |
| PS | BIPARTITE GRAPH (NEUROCORE) | 0.852 | 0.749 | 0.721 | 0.774 | 0.920 | 0.809 | 0.766 | 0.832 | 0.826 | 0.826 | 0.835 | 0.829 |
| | PASAT (HG) | 0.855 | 0.756 | 0.739 | 0.784 | 0.923 | 0.818 | 0.788 | 0.843 | 0.823 | 0.829 | 0.863 | 0.838 |
| | PASAT (HG+DE) | 0.859 | 0.767 | 0.750 | 0.792 | 0.924 | 0.828 | 0.800 | 0.851 | 0.832 | 0.851 | 0.876 | 0.853 |
| | PASAT (HG+DE+REG, FULL) | 0.864 | 0.775 | 0.752 | 0.797 | 0.930 | 0.838 | 0.803 | 0.857 | 0.845 | 0.861 | 0.880 | 0.862 |

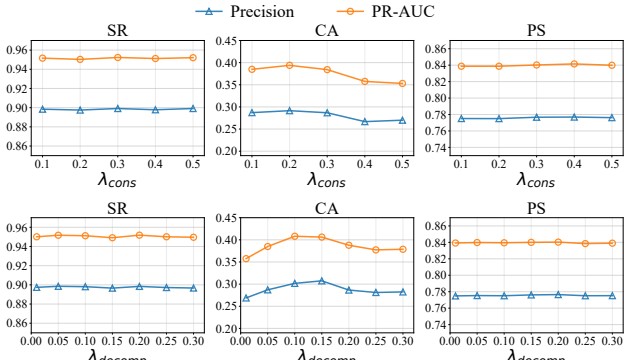

*Figure 2.* Parameter Analysis.

(HG), PASAT (HG+DE), and PASAT (HG+DE+REG). Results are reported in Table 2.

Across the three datasets, we incrementally build upon a bipartite-graph baseline by adding the components of PASAT, which results in consistent performance improvements on most difficulty splits. On the SR and PS datasets, hypergraph-based message passing already improves performance over the baseline, and the full model further increases Precision by up to 2.5% and 3.0%, respectively. On the CA dataset, the benefits of incremental integration are more pronounced: hypergraph modeling improves the average Precision from 0.238 to 0.360, invariant–equivariant variable decomposition further raises it to 0.383, and the full model reaches 0.408, corresponding to a 71.4% improvement over the baseline. The gains are particularly evident on the Medium and Hard splits, indicating that PASAT effectively captures complex structural patterns and learns from challenging instances with sparse unsat-core variables. Overall, these results demonstrate that each component contributes incrementally to performance improvements.

**Exp-3: Parameter Analysis.** We analyze two regularization terms, $\lambda_{cons}$ and $\lambda_{decomp}$. We fix $\lambda_{decomp} = 0.05$ and vary $\lambda_{cons}$ within $[0.1, 0.5]$ with a step size of $0.1$. We then fix $\lambda_{cons} = 0.1$ and vary $\lambda_{decomp}$ within $[0.01, 0.30]$ with a step size of $0.05$. All other settings are the same as in Exp-1. Performance is evaluated on the Medium split of

*Table 3.* Solving results on SAT Competition instances.

| DATASET | SOLVER | #SOLVED | #SAT | #UNSAT |
|---|---|---|---|---|
| SATCOMP 2023 | GLUCOSE-DEFAULT | 133 | 59 | 74 |
| | GLUCOSE-PASAT | 148 | 64 | 84 |
| SATCOMP 2024 | GLUCOSE-DEFAULT | 111 | 50 | 61 |
| | GLUCOSE-PASAT | 118 | 55 | 63 |
| SATCOMP 2025 | GLUCOSE-DEFAULT | 135 | 60 | 75 |
| | GLUCOSE-PASAT | 141 | 62 | 79 |

each dataset. The results are reported in Figure 2. On the SR and PS datasets, model performance remains relatively stable across different values of both $\lambda_{cons}$ and $\lambda_{decomp}$, indicating low sensitivity to the regularization strength. These datasets exhibit relatively simple structural characteristics, and applying a small amount of regularization is sufficient to obtain strong performance. In contrast, the CA dataset shows clearer performance trends with respect to the two regularization terms. Performance improves when $\lambda_{cons}$ lies in $[0.1, 0.3]$ and $\lambda_{decomp}$ lies in $[0.05, 0.2]$. The CA dataset involves more complex variable–clause structures and is more difficult for representation learning, leading to overall lower performance and greater sensitivity to hyperparameter choices. Moreover, larger regularization weights may encourage the model to focus more on polarity-related information, thereby weakening the supervision from the main task and resulting in performance degradation.

**Exp-4: Effectiveness of Combination with SAT Solvers.** We evaluate the performance of a CDCL SAT solver combined with our neural model. Specifically, we modify the Glucose solver (Audemard & Simon, 2017) by incorporating the predicted variable scores as additional inputs to guide the solving process. The neural model is trained on instances from the SAT Competition Main Track [1] between 2017 and 2022, following the same training procedure as NeuroCore (Selsam & Bjørner, 2019). Training details are provided in Appendix B.2. The trained model is then used to generate variable scores for instances from the SATCOMP Main Track in 2023-2025, each of which contains 400 instances. We compare the modified solver with the original

---

[1] https://satcompetition.github.io/

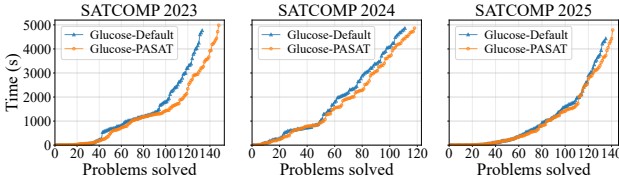

*Figure 3.* Solving times for SAT Competition instances.

Glucose solver under identical configurations. All instances are solved under a time limit of 5000 seconds. The numbers of solved instances are reported in Table 3. On the SAT-COMP datasets, Glucose augmented with our method solves more instances overall, outperforming the original Glucose solver by 15, 7, and 6 instances, respectively. Improvements are observed on both SAT and UNSAT instances, indicating that the proposed method provides effective variable-level information for CDCL solving.

In addition, we report cactus plots to illustrate the solving process in Figure 3. Across all datasets, Glucose-PASAT outperforms the default Glucose solver overall. While the two configurations exhibit similar performance on easier instances, Glucose-PASAT demonstrates clear advantages in the medium-to-hard range, where the solving-time curve increases more slowly. These results indicate that the proposed polarity-aware neural guidance provides effective variable-level information that helps the solver navigate difficult search spaces more efficiently. Additional solver-integration results are provided in Appendix C.3.

## 5. Related Work

In this section, we review related studies on learning-based approaches for SAT solving.

**End-to-End Learning for SAT Solving.** A number of studies employ neural networks to solve SAT problems directly (Bünz & Lamm, 2017; Selsam et al., 2019; Amizadeh et al., 2019; Li et al., 2023; Cameron et al., 2020; Chen et al., 2025), most of which formulate SAT solving as a single-bit prediction task for satisfiability. Early work (Bünz & Lamm, 2017) models CNF formulas as bipartite graphs of literals and clauses and applies GNNs for satisfiability classification. NeuroSAT (Selsam et al., 2019) extends this representation by connecting complementary literals and performing message passing between literals and clauses. DG-DAGRNN (Amizadeh et al., 2019) targets circuit SAT by modeling instances as directed acyclic graphs and learning satisfying assignments with gated recursive networks. DeepSAT (Li et al., 2023) represents SAT instances as AND-inverter graphs and leverages polarity-aware DAG neural networks to model Boolean constraint propagation. Other studies investigate permutation-invariant architectures for learning directly from CNF formulas (Cameron et al., 2020) or unsupervised end-to-end frameworks for Weighted MaxSAT based on hypergraph neural networks (Chen et al.,

2025). In contrast to these approaches, we focus on unsat-core prediction, rather than directly predicting the satisfiability of SAT instances. We model CNF formulas as clause–literal hypergraphs augmented with a clause–clause incidence graph, which enables the capture of higher-order relationships among literals and clauses.

**Learning-Guided SAT Solving.** Another line of research leverages neural networks to guide heuristics in SAT solvers, aiming to reduce the reliance on domain expertise. Most existing methods focus on CDCL solvers, with some extensions to stochastic local search (SLS). For CDCL solvers, (Liang et al., 2018) propose a learning-based restart policy that predicts the quality of learned clauses. Neuro-Core (Selsam & Bjørner, 2019) extends NeuroSAT to predict unsat-core variables and guides variable selection accordingly. Graph-Q-SAT (Kurin et al., 2020) introduces a reinforcement learning–based branching heuristic, while NeuroBack (Wang et al., 2024) improves phase selection by predicting backbone variable phases prior to solving. NeuroSelect (Liu et al., 2024) learns adaptive clause-deletion strategies using a graph transformer, and RLAF (Tönshoff & Grohe, 2025) injects learned variable weights and polarities into branching heuristics in a one-shot manner. For SLS solvers, (Yolcu & Póczos, 2019) and NLocalSAT (Zhang et al., 2020) apply GNNs to variable selection and assignment initialization, respectively, while NSNet (Li & Si, 2022) formulates SAT and #SAT as probabilistic inference and applies GNN-based belief propagation. In addition, GraSS (Zhang et al., 2024) addresses solver selection via graph-based representations. Our work follows the learning-guided paradigm for CDCL solvers by predicting unsat-core variables, and further enhances it with decomposed message passing and polarity-aware consistency regularization.

## 6. Conclusion

In this paper, we presented a polarity-aware hypergraph-based framework for unsat-core variable prediction in SAT problems. By modeling SAT formulas as clause–literal hypergraphs augmented with clause-level interactions, the proposed approach captures higher-order structural dependencies. The introduction of invariant–equivariant variable decomposition enables explicit modeling of the relationship between positive and negative literals, while the polarity-inversion consistency regularization further enforces intrinsic symmetry properties of SAT during training.

Experimental results across multiple datasets demonstrate that PASAT consistently improves unsat-core variable prediction over existing learning-based approaches, and offers practical benefits when integrated with CDCL solvers. Future work includes developing more expressive hypergraph-based representations for SAT formulas and extending the framework to a broader range of SAT problems.

## Acknowledgements

We thank the anonymous reviewers for their constructive comments and suggestions. This work was supported by the National Natural Science Foundation of China (Grant No. U22B2021, U24B20143, and 62572034), State Key Laboratory of Complex & Critical Software Environment (SKLCCSE), and CIE–Tianyi Cloud Research Program.

## Impact Statement

This paper presents work whose goal is to advance the field of machine learning. There are many potential societal consequences of our work, none of which we feel must be specifically highlighted here.

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

## A. Dataset Details

In unsat-core prediction tasks, we use three generated SAT datasets, namely SR, Community Attachment (CA), and Popularity-Similarity (PS), following the dataset construction procedure of G4SATBench (Li et al., 2024). Specifically, CNF formulas are generated using the SR generator from NeuroSAT (Selsam et al., 2019), the Community Attachment (CA) model (Giráldez-Cru & Levy, 2015), and the Popularity-Similarity (PS) model (Giráldez-Cru & Levy, 2017). Each dataset is divided into three difficulty levels: easy, medium, and hard. For each difficulty level, the training set contains 80,000 pairs of satisfiable and unsatisfiable instances, while the validation and test sets each contain 10,000 such pairs. For the unsat-core prediction task, we use only unsatisfiable instances for both training and evaluation, following the same setting as NeuroSAT (Selsam et al., 2019). The detailed statistics of these datasets are summarized in Table 4.

*Table 4.* Statistics of datasets with difficulty splits.

| DATASET | SPLIT | VARIABLES | | | CLAUSES | | | UNSAT-CORE VARS | | |
|---|---|---|---|---|---|---|---|---|---|---|
| | | AVG. | MIN | MAX | AVG. | MIN | MAX | AVG. | MIN | MAX |
| SR | EASY | 24.99 | 10 | 40 | 148.32 | 23 | 355 | 19.64 | 2 | 40 |
| | MEDIUM | 120.11 | 40 | 200 | 654.81 | 128 | 1332 | 99.49 | 2 | 200 |
| | HARD | 300.10 | 200 | 400 | 1616.47 | 420 | 2449 | 277.83 | 3 | 400 |
| CA | EASY | 30.39 | 16 | 40 | 278.67 | 57 | 590 | 5.74 | 4 | 40 |
| | MEDIUM | 119.60 | 40 | 200 | 1651.41 | 162 | 2999 | 17.16 | 4 | 118 |
| | HARD | 299.93 | 200 | 400 | 4197.51 | 2601 | 5998 | 42.19 | 19 | 146 |
| PS | EASY | 27.25 | 10 | 40 | 192.22 | 54 | 319 | 18.86 | 3 | 40 |
| | MEDIUM | 123.48 | 40 | 200 | 869.65 | 224 | 1599 | 65.86 | 3 | 200 |
| | HARD | 250.17 | 200 | 300 | 1761.98 | 917 | 2399 | 127.02 | 3 | 300 |

In Exp-4, we construct the training data using instances from the SAT Competition Main Track spanning 2017 to 2022, following the data generation procedure for unsat core variable prediction described in (Selsam & Bjørner, 2019). Instances from the SAT Competition Main Track in 2023 to 2025 are reserved exclusively for solver evaluation. During evaluation, the trained model predicts variable scores for each instance, which are used to guide variable selection in the Glucose solver. Statistics of the training and evaluation datasets are summarized in Table 5.

*Table 5.* Statistics of SAT Competition Main Track instances.

| YEAR | #INST. | STATUS | | VARIABLES | | | | CLAUSES | | | |
|---|---|---|---|---|---|---|---|---|---|---|---|
| | | SAT | UNSAT | AVG. | MED. | MIN | MAX | AVG. | MED. | MIN | MAX |
| SATCOMP 2017 | 350 | 133 | 140 | 521955.61 | 7411.5 | 180 | 8905808 | 1879089.81 | 138764.0 | 648 | 32322587 |
| SATCOMP 2018 | 400 | 192 | 134 | 360371.86 | 29690.0 | 175 | 3171415 | 1646869.57 | 248471.0 | 1119 | 17708937 |
| SATCOMP 2019 | 399 | 161 | 128 | 247171.81 | 8591.0 | 97 | 9582411 | 1562906.69 | 117206.0 | 204 | 103690720 |
| SATCOMP 2020 | 400 | 191 | 145 | 310404.10 | 30999.5 | 54 | 8043699 | 4007261.77 | 274643.0 | 58 | 129333040 |
| SATCOMP 2021 | 400 | 156 | 187 | 192789.42 | 24062.0 | 44 | 3416996 | 2733968.29 | 236609.5 | 513 | 67009046 |
| SATCOMP 2022 | 400 | 172 | 186 | 981553.01 | 34939.0 | 99 | 18707849 | 7117471.09 | 292902.0 | 264 | 214309011 |
| SATCOMP 2023 | 400 | 154 | 219 | 911146.56 | 11636.5 | 45 | 21185701 | 4477681.16 | 81176.5 | 376 | 96368706 |
| SATCOMP 2024 | 400 | 179 | 208 | 1435523.72 | 4566.5 | 74 | 48505464 | 5028755.65 | 62239.5 | 252 | 130975382 |
| SATCOMP 2025 | 400 | 96 | 80 | 1588632.13 | 24367.0 | 149 | 117295030 | 6015936.93 | 259340.0 | 606 | 315573317 |

## B. Detailed Experimental Settings

### B.1. Implementation Details and Hyperparameters

All methods are implemented using PyTorch[2]. The experiments are conducted on a workstation equipped with an Intel Xeon Gold 6148 CPU at 2.40 GHz and an NVIDIA Tesla V100 PCIe GPU with 32 GB memory. The GPU is used for training and evaluating neural models, while the CPU is used for data processing and solving SAT instances with classical solvers. The hyperparameter settings used in our experiments are summarized in Table 6. Code available at https://github.com/sunzc-super/PASAT-ICML.

---

[2] https://pytorch.org/

*Table 6.* Hyperparameter settings used in our experiments.

| HYPERPARAMETER | VALUE RANGE / SET |
|---|---|
| HIDDEN DIMENSION | 80 |
| MESSAGE PASSING ROUNDS ($T$) | $\{2, 4\}$ |
| MLP LAYERS | $\{2, 3\}$ |
| EPOCHS | $\{100, 200\}$ |
| BATCH SIZE | $\{100, 200\}$ |
| LEARNING RATE | $\{$1E-4, 2E-4, 4E-4$\}$ |
| WEIGHT DECAY | $\{$1E-4, 5E-4$\}$ |
| GRADIENT CLIPPING | $\{1, 2, 5, 10\}$ |
| LR SCHEDULER | EXPONENTIALLR |
| LR DECAY FACTOR ($\gamma$) | $\{0.871, 0.95\}$ |
| CONSISTENCY WEIGHT ($\lambda_{\text{cons}}$) | $\{0.1, 0.3\}$ |
| DECOMPOSITION WEIGHT ($\lambda_{\text{decomp}}$) | $\{0.05, 0.15\}$ |

### B.2. SAT Solver Integration Details

In Exp-4, we follow the training and evaluation pipeline proposed in NeuroCore (Selsam & Bjørner, 2019). We first generate training data using SAT Competition Main Track instances from 2017 to 2022, following the same data construction procedure. For each year, we generate 20,000 training samples. The resulting dataset is used to train our neural model for unsat core variable prediction. The training settings are the same as in the previous experiments. After training, the model is applied to SAT Competition Main Track instances from 2023 to 2025 to predict unsat core variable scores for each instance.

In contrast to NeuroCore, we modify the Glucose solver to accept both the CNF formula and the predicted variable scores as input. We use Glucose release 4.2.1 as the base solver. During solving, each instance is limited to a time budget of 5000 seconds. The modified solver periodically updates variable activity scores every 100 seconds by overwriting them with the predicted variable scores, thereby guiding variable selection during solving. To satisfy GPU memory constraints during inference, we restrict the maximum problem size processed by the neural model. Specifically, the total number of literals and variables is limited to at most 300,000, and the number of literal–clause incidence relations is limited to at most 2,000,000, similar to the data limits used in NeuroCore. In addition, when constructing the clause–clause graph, we retain only the top-50 weighted edges for each clause, corresponding to the strongest inter-clause relationships.

## C. Additional Experimental Results

### C.1. Unsat-Core Prediction Results on SATCOMP Datasets

To evaluate the generalization performance on industrial-scale instances, we additionally report the unsat-core prediction results on the SATCOMP 2023 and 2024 datasets, which include structured instances from real-world domains. Compared with synthetic benchmarks, these instances are typically more diverse in structure and closer to practical SAT applications, thereby providing an additional test of whether the learned representations can generalize beyond synthetic datasets. The models were trained following the protocol described in Exp-4 and Appendix B.2. For evaluation, we constructed the test sets from SATCOMP 2023 and 2024 by first extracting CNF–unsat-core pairs during preprocessing and then randomly sampling 800 pairs from each year. The results are reported in Table 7. PASAT achieves better performance than NeuroCore on both datasets. These improvements indicate that the proposed clause–literal hypergraph augmented with a clause incidence graph, together with polarity-aware decomposition and consistency regularization, can learn effective literal representations on more realistic and structurally diverse SAT instances. While this evaluation does not fully cover all industrial-scale settings, it provides additional evidence that PASAT can generalize beyond the synthetic SR, CA, and PS datasets.

*Table 7.* Unsat-core prediction performance on SATCOMP 2023 and SATCOMP 2024.

| METHOD | SATCOMP 2023 | | | SATCOMP 2024 | | |
|---|---|---|---|---|---|---|
| | PRECISION | PR-AUC | ROC-AUC | PRECISION | PR-AUC | ROC-AUC |
| NEUROCORE | 0.699 | 0.705 | 0.779 | 0.724 | 0.743 | 0.784 |
| PASAT | **0.705** | **0.712** | **0.803** | **0.754** | **0.761** | **0.853** |

## C.2. Inference Time and Memory Usage

To evaluate the runtime overhead of PASAT, we report the average inference time per instance and the peak GPU memory usage during testing on the SR, CA, and PS datasets. The results are reported in Table 8. We compare PASAT with NeuroCore under the same testing setting across Easy, Medium, and Hard splits.

Overall, PASAT incurs slightly higher inference time and GPU memory usage than NeuroCore in most cases. However, the average inference time remains at the millisecond level, indicating that the additional computational overhead is limited. The peak GPU memory usage of PASAT also remains close to that of NeuroCore across different datasets and difficulty levels. These results show that the performance improvements of PASAT are achieved with a practical inference overhead.

*Table 8.* Runtime and memory comparison between NeuroCore and PASAT during testing. Avg. infer time denotes the average inference time per instance in milliseconds. Peak VRAM denotes the peak GPU memory usage in MB.

| Split | Dataset | NeuroCore | | PASAT | |
|---|---|---|---|---|---|
| | | Avg. Infer Time (ms) | Peak VRAM (MB) | Avg. Infer Time (ms) | Peak VRAM (MB) |
| Easy | SR | 2.61 | 161.48 | 2.89 | 189.58 |
| | CA | 3.62 | 302.59 | 4.12 | 315.28 |
| | PS | 3.30 | 207.15 | 3.91 | 236.24 |
| | Avg. | 3.18 | 223.74 | 3.64 | 247.03 |
| Medium | SR | 4.06 | 710.06 | 3.53 | 835.28 |
| | CA | 4.16 | 1679.78 | 6.31 | 1730.22 |
| | PS | 4.02 | 964.36 | 4.92 | 1084.76 |
| | Avg. | 4.08 | 1118.07 | 4.92 | 1216.75 |
| Hard | SR | 4.33 | 1706.10 | 5.69 | 1985.87 |
| | CA | 3.88 | 4139.99 | 14.43 | 4160.48 |
| | PS | 3.95 | 1912.54 | 11.54 | 2089.94 |
| | Avg. | 4.05 | 2586.21 | 10.55 | 2745.43 |

## C.3. Solver Integration with CaDiCaL

To further evaluate whether PASAT can improve modern CDCL solvers beyond Glucose, we additionally integrate PASAT with CaDiCaL [3]. Specifically, we use CaDiCaL 1.9.5 and run both CaDiCaL and CaDiCaL-PASAT in plain mode to better isolate the effect of neural-guided variable scoring. We report the number of solved instances on SATCOMP 2023, 2024, and 2025 under the same evaluation setting as in Exp-4 and Appendix B.2.

The results are reported in Table 9. CaDiCaL-PASAT consistently solves more instances than the original CaDiCaL across all three SATCOMP benchmarks. In particular, CaDiCaL-PASAT solves 2, 4, and 6 more instances on SATCOMP 2023, 2024, and 2025, respectively. These results indicate that the learned variable scores produced by PASAT can provide useful guidance for CaDiCaL and further support the effectiveness of PASAT in neural-guided SAT solving.

*Table 9.* Solver integration results with CaDiCaL on SATCOMP 2023, 2024, and 2025.

| Dataset | Solver | #Solved | #SAT | #UNSAT |
|---|---|---|---|---|
| SATCOMP 2023 | CaDiCaL | 236 | 116 | 120 |
| | CaDiCaL-PASAT | **238** | **117** | **121** |
| SATCOMP 2024 | CaDiCaL | 252 | 122 | 130 |
| | CaDiCaL-PASAT | **256** | **126** | 130 |
| SATCOMP 2025 | CaDiCaL | 240 | 128 | 112 |
| | CaDiCaL-PASAT | **246** | **132** | **114** |

---

[3] https://github.com/arminbiere/cadical

