# OpenReview forum: "Unsat Core Prediction through Polarity-Aware Representation Learning over Clause-Literal Hypergraphs"
_ICML.cc/2026/Conference — ICML 2026 regular_

### Official Review · Reviewer_miRB · 2026-02-13

**Soundness:** 3
**Presentation:** 3
**Significance:** 3
**Originality:** 3
**Overall Recommendation:** 5
**Confidence:** 5

**Summary:**

This work aims to learn a core-prediction model for SAT solver guidance by aiming to structure the architecture and the graph representation to more directly capture the constrainedness of literals by their inverse polarities, and of clauses by their incident counterparts.

**Compliance With Llm Reviewing Policy:**

Affirmed.

**Final Justification:**

In the 2nd rebuttal, my remaining concerns were addressed adequately. To reflect this, I have raised my score from a 4 to a 5.

**Key Questions For Authors:**

Q1. Can you please address the following list of vague/inaccurate claims:
	1. Abstract: “However, most existing approaches model a SAT formula as a bipartite graph or a directed acyclic graph, which are less expressive in capturing higher-order interactions”. Less expressive than what?
	2. L. 063: “they do not explicitly enforce two key constraints: (i) complementary literals of the same variable should share common information”. In a message-passing framework, adjacent nodes necessarily share information, since they are each a function of eachother’s previous states.

Q2: I think you are missing several related works. A search for “unsat core prediction” in Google Scholar returns several works which are not included in your discussion:
* Abdulla, P. A., Liang, C., & Rümmer, P. (2023, December). Boosting constrained Horn solving by unsat core learning. In International Conference on Verification, Model Checking, and Abstract Interpretation (pp. 280-302). Cham: Springer Nature Switzerland.
* Cotnareanu, J., Zhang, Z., Zhen, H.L., Zhang, Y. and Coates, M., 2024. HardCore Generation: Generating Hard UNSAT Problems for Data Augmentation. Advances in Neural Information Processing Systems, 37, pp.62409-62431.
* Chen, G. and Wang, J., 2025, June. Enhancing Modern SAT Solver With Machine Learning Method. In Proceedings of the Great Lakes Symposium on VLSI 2025 (pp. 886-892).

For each of these (and for others which you might find), I think you need to either explain why you can or should not compare against these as baselines, or do so. For example, the “Boosting constrained Horn solving […]” paper may be somewhat out of domain, to the point where it may be worth mentioning as related work but too difficult to adapt to the setting to be worth it. However, the other two papers seem well-grounded in your field and both report core-classification accuracy, implying that their methods can both output the predictions you use for solver guidance.

Q3. In what way is your hypergraph construction actually different form the literal-clause graph construction commonly used. Your described incidence matrix of the hypergdaph, $H \in R^{2N \times M}$, is exactly the adjacency matrix one would get by constructing the bipartite LCG. In this case, a bipartite message-massing algorithm would make all the same computations as your described message passing algorithm. I don’t mind you calling the graph a hypergraph instead of the LCG, but I don’t think it is accurate to claim that it is more generalized or informative than the LCG. Considering that your ablation shows that HG alone makes no difference in performance, I think you may have difficulty demonstrating that this is not the case.

Q4. In the LCG, each clause node is 2 message-passes (hops) away from the clause nodes which it is 1 hop away from in the CIG. So, the LCG captures the same inter-clause dependencies as the CIG, with the limitation of requiring twice the hops. If the argument is that the CIG allows you to keep the number of GNN layers lower to avoid smoothing (as you allude to in l. 083), that’s fine but your text at line 140 needs changing then. Also, considering the bi-partite graph is fairly sparse (in particular with respect to SAT problems) relative to the standard connected graph, I’m not convinced that smoothing occurs as rapidly in these graphs as you imply.

Q5: I understand that the KL divergence loss is consistent with NeuroCore, but why not simply use a binary cross-entropy loss? You’re doing binary node classification, which typically has a standardized training framework.

Q6: You eventually train your model on SAT Competition data. Why not present model accuracy on that data, as you do for your synthetic data?

Q7: You say you follow the NeuroCore procedure for combining your model with Glucose, so why not include NeuroCore in the comparison in Table 3?

**Limitations:**

Yes

**Strengths And Weaknesses:**

Strengths
S1. The paper is generally clear and well written
S2. Experimental procedure seems sound

Weaknesses
W1. The paper makes some unsupported and/or vague claims
W2. The work seems to exclude some relevant prior work in both discussion and experimental sections

---

> ### Author Rebuttal · Authors · 2026-03-31
>
> We sincerely thank you for the detailed suggestions and for the careful and thorough review of our work. Below we provide our response to the reviewer's insightful comments.
>
> > **Q1: address the claim: (1) "... as a bipartite graph or a directed acyclic graph, which are less expressive in capturing higher-order interactions". Less expressive than what? \
> (2) “they do not explicitly enforce two key constraints: ... ”. In a message-passing framework, adjacent nodes necessarily share information, since they are each a function of each other’s previous states.**
>
> Thanks for your valuable comments.
>
> (1) We will revise the term “less expressive” for clarity.
> For "expressive in capturing higher-order interactions," we meant that the LCG requires two-hop propagation via literal nodes to capture clause-level relations, whereas our HG with the CIG makes such interactions one-hop accessible, as you noted in Q3, rather than that such dependencies cannot be captured at all.
>
> (2) We agree that message passing enables information exchange between adjacent nodes. Our point is not that complementary literals cannot interact, but that existing frameworks do not explicitly enforce what should be shared between them or how they should behave under polarity inversion. Our decomposition and consistency regularization are introduced for this purpose.
>
> We will clarify these points in the revised paper.
>
> > **Q2: Missing several related works.**
>
> Thank you for your valuable suggestion.
>
> As you pointed out, [R1] is somewhat outside the main scope of our work, since it is designed for constrained Horn clause solving. However, its clause-level core prediction idea is still relevant and informative for our study.
>
> [R2] focuses on SAT instance generation rather than unsat-core prediction. While it includes a core prediction module, this module is built upon GCN and is therefore closely related to the GCN baseline in our experiments.
>
> For [R3], we additionally implemented its method, denoted as WLIG-GNN, and report the results on the medium splits of SR/CA/PS.
>
> |Method|Precision (SR)|PR-AUC (SR)|ROC-AUC (SR)|Precision (CA)|PR-AUC (CA)|ROC-AUC (CA)|Precision (PS)|PR-AUC (PS)|ROC-AUC (PS)|
> |-|-:|-:|-:|--:|--:|-:|-:|-:|-:|
> |WLIG-GNN|0.869|0.915|0.770|0.180|0.266|0.559|0.730|0.789|0.798|
> |PASAT|0.900|0.953|0.863|0.307|0.406|0.666|0.775|0.838|0.861|
>
> These results show that PASAT consistently outperforms WLIG-GNN across all three datasets. We also report additional baseline results under W1 for reviewer 7qnW.
> We will include the above results and add more discussion of related work in the revised paper.
>
> [R1] Boosting Constrained Horn Solving by Unsat Core Learning. In VMCAI, 2024. \
> [R2] HardCore Generation: Generating Hard UNSAT Problems for Data Augmentation. In NeurIPS, 2024. \
> [R3] Enhancing Modern SAT Solver With Machine Learning Method. ACM Great Lakes Symposium on VLSI, 2025.
>
> > **Q3 : In what way is your hypergraph actually different form the LCG commonly used.**
>
> Thanks for your insightful comment. We agree that the HG and the LCG have the same incidence structure.
>
> The difference mainly lies in the modeling formulation. In the LCG, both literals and clauses are modeled as nodes, and their relations are represented by edges, whereas in the HG formulation, literals are represented as nodes and clauses as hyperedges.
> In addition, we further introduce the CIG, which makes clause-level interactions one-hop accessible.
>
> We will clarify this more carefully in the revised paper.
>
> > **Q4. ... the LCG captures the same inter-clause dependencies as the CIG, with the limitation of requiring twice the hops. .... I’m not convinced that smoothing occurs as rapidly in these graphs as you imply.**
>
> Thank you for your insightful comments.
> As you noted, related clauses can interact in the LCG through 2-hop propagation via literal nodes, whereas the CIG makes such dependencies one-hop accessible.
> By shortening propagation paths, the CIG may reduce the need for deeper message passing and hence lower the potential risk of over-smoothing, without implying that it necessarily occurs in SAT graphs.
> We will clarify this point and adjust the wording accordingly in the revised paper.
>
> > **Q5: why not use a binary cross-entropy loss?**
>
> Thanks for your valuable comment. We use KL divergence because the supervision is not strictly binary. For one UNSAT instance, multiple valid unsat cores may exist. Thus, binary cross-entropy may enforce overly strict supervision on one specific core, whereas KL divergence allows the model to learn a softer relevance distribution. This choice is also consistent with prior work such as NeuroCore and SATFormer.
>
> > **Q6: Why not present model accuracy on SAT Competition data?**
>
> Please kindly refer to the response to **W2 of reviewer GWMK** due to the limitation of space.
>
> > **Q7: Why not include NeuroCore in the comparison?**
>
> Please kindly refer to the response to **W2 of reviewer RKoq** due to the limitation of space.

---

> > ### Author Rebuttal · Reviewer_miRB · 2026-03-31
> >
> > Q1.
> >
> > The one-hop vs. multi-hop argument is not sufficient to justify using a hypergraph, given the ablation study results showing that the hyper-graph has no real change compared to NeuroCore in prediction performance. Your argument towards the regularization elements is more convincing.
> >
> > Q2. OK.
> >
> > Q3. Again, I don't see why it matters that the nodes are one-hop accessible. The convincing response to this would have been to demonstrate that the LCG does in fact at faster than twice the rate of the HG (since each 2 LCG hops is 1 LCG hop), to show that as you increase the number of MP layers the LCG-based GCN architecture caps out in performance before the HG-based one does (again accounting for the 2:1 hop conversion), or to ablate your method such that each component of your method is present except for the HG, and that without the HG the other components of your method are not useful.
> >
> > Q4. This is effectively a non-claim. While your revision will be accurate, it will be of little impact.
> >
> > Q5. I had presumed that you use only the minimal core as a ground-truth label. If you accept any core, and are presumably measuring your classification metrics per-node, how do you know which of the acceptable cores the model is targetting? Practically, is your ground-truth label just the union of all acceptable cores? What is the average percentage of variables which are labelled as positive classifications?
> >
> > Q6. OK.
> >
> > Q7: OK. These results show pretty marginal improvement over the NeuroCore baseline.

---

> > > ### Author Response · Authors · 2026-04-04
> > >
> > > Thank you sincerely for your time, constructive feedback, and positive recognition of our rebuttal.
> > >
> > > > **Q1. The one-hop vs. multi-hop argument is not sufficient to justify using a hypergraph, given the ablation study results showing that the hyper-graph has no real change compared to NeuroCore in prediction performance.**
> > >
> > > Thank you for your great comment! We would like to clarify that “PASAT (HG)” in our ablation study actually refers to the combined HG+CIG component rather than the hypergraph alone.
> > >
> > > Your observation is very insightful. The ablation results on SR show limited improvement because literals in each clause are randomly sampled from a predefined set of N variables, leading to weak literal-clause structural patterns. In contrast, the ablation results on the PS and CA datasets still indicate that the **combined HG+CIG design** can bring clear gains over NeuroCore, as these datasets are constructed with certain literal-clause constraints and thus exhibit more structured literal-clause interaction patterns, reflecting popularity-similarity in PS and community structure in CA.
> > >
> > > For example, Precision improves **from 0.749 to 0.756** and **from 0.721 to 0.739** on the medium and hard splits of PS, respectively, and **from 0.191 to 0.263** and **from 0.189 to 0.482** on the medium and hard splits of CA, respectively. The improvement is also more pronounced on the more difficult splits.
> > >
> > > We hope the above clarification helps alleviate your concern.
> > >
> > >
> > > > **Q3. I don't see why it matters that the nodes are one-hop accessible...** \
> > > > **Q4. This is effectively a non-claim.**
> > >
> > > Thank you for this helpful comment! The more direct modeling of clause-level interactions is introduced by the **additional CIG** rather than by the HG alone.
> > >
> > > The intuition of clause-level interactions, or making such interactions more directly accessible, is also consistent with several related SAT-oriented studies. For example, NeuroBack introduces an additional meta-node to connect clause nodes within the same connected components [R1], thereby shortening propagation paths in the graph. SATformer uses a hierarchical Transformer-based architecture to directly capture clause correlations [R2]. Furthermore, a recent study on hypergraphs combines the line graph, which is similar in spirit to our CIG, to effectively capture interactions among hyperedges [R3].
> > >
> > > We will clarify this point more carefully in the revised paper.
> > >
> > > [R1] NeuroBack: Improving CDCL SAT Solving using Graph Neural Networks. In ICLR, 2024. \
> > > [R2] SATformer: Transformer-Based UNSAT Core Learning. In ICCAD, 2023. \
> > > [R3] Hypergraph-enhanced Dual Semi-supervised Graph Classification. In ICML, 2024.
> > >
> > > > **Q5. Unsat-Core Label Construction.**
> > >
> > > Thanks for your valuable comment!
> > >
> > > For the SR/CA/PS datasets, we follow G4SATBench and use CaDiCaL with DRAT-trim to extract unsat-core variables. Each instance is associated with one extracted core, whose variables serve as ground-truth labels. Detailed statistics are provided in Appendix A.1.
> > >
> > > For the SATCOMP datasets, following NeuroCore, we use Z3 with DRAT-trim to extract unsat cores. For each UNSAT instance, the code first stores one core and then attempts to find up to 10 additional ones, so a single instance may yield multiple CNF--unsat-core pairs. The average percentage of variables labelled as positive is 61.60%.
> > >
> > > During both training and evaluation, we treat each CNF--unsat-core pair as a separate sample and compute the prediction metrics against its corresponding labels.
> > > In addition, the idea of considering the union of all acceptable cores is insightful, and we believe it could be a meaningful direction for future study.
> > >
> > > We will clarify these points more explicitly in the revised paper.

---

### Official Review · Reviewer_RKoq · 2026-03-05

**Soundness:** 3
**Presentation:** 3
**Significance:** 3
**Originality:** 3
**Overall Recommendation:** 4
**Confidence:** 4

**Summary:**

The paper introduces PASAT, a polarity-aware representation learning framework designed to improve unsat-core variable prediction in SAT problems. To overcome the limitations of traditional bipartite graph models, the authors represent SAT formulas as clause-literal hypergraphs augmented with a clause incidence graph, effectively capturing higher-order interactions among literals and clauses. A primary contribution is the polarity-aware decomposed message-passing mechanism, which separates variable representations into invariant and equivariant components to explicitly model the complementary relationship between positive and negative literals. Furthermore, the framework employs a polarity-inversion consistency regularization during training.

**Compliance With Llm Reviewing Policy:**

Affirmed.

**Final Justification:**

The rebuttal has thoroughly address my main concerns. I will maintain my positive score of 4.

**Key Questions For Authors:**

1. Have you evaluated, or could you provide during the rebuttal period, the performance of PASAT when integrated with a modern state-of-the-art CDCL solver, such as Kissat, rather than relying solely on Glucose?

2. In the end-to-end solver experiments, you compare Glucose-PASAT strictly against Glucose-Default. Can you provide an end-to-end comparison against a solver integrated with a prior neural guidance method, such as Glucose-NeuroCore?

**Limitations:**

Please refer to the weaknesses above

**Strengths And Weaknesses:**

# Strengths
1. Enhancing SAT solvers via machine learning addresses a critical problem with broad implications for fields like planning, verification, and security. The paper effectively demonstrates that PASAT consistently outperforms established baselines, such as NeuroCore and SATFormer, in unsat-core prediction across multiple difficulty levels. This pushes the boundaries of what Graph Neural Networks can extract from SAT formulas.
2. The introduction of the invariant-equivariant variable decomposition is a highly creative and domain-specific contribution. It addresses a fundamental property of SAT, that complementary literals share a root variable but possess inverted polarities. This is a significant step forward from prior works, which largely overlooked this intrinsic relationship by simply concatenating literal embeddings.

# Weaknesses
1. The end-to-end solving evaluation is exclusively restricted to the Glucose solver. While Glucose is a solid foundational CDCL solver, it does not represent the current state-of-the-art. To convincingly demonstrate the method's practical utility in modern competitive environments, integrating the neural model with a contemporary SOTA solver, such as Kissat, is necessary.
2. In the solver experiments, Glucose-PASAT is only compared against the default Glucose solver. To rigorously prove the advantage of the PASAT architecture over prior neural guidance methods, the authors must include a comparison against a baseline like Glucose-NeuroCore.
3. The application of hypergraphs for SAT representation is not entirely unprecedented. However, the paper successfully mitigates this by coupling the structural representation with their novel polarity-aware decomposition mechanisms.

---

> ### Author Rebuttal · Authors · 2026-03-31
>
> We sincerely thank you for this highly positive comment and for recognizing the novelty of our invariant-equivariant variable decomposition as a creative and domain-specific contribution. Below we provide our response to your insightful comments.
>
> > **W1 & Q1: Have you evaluated, or could you provide during the rebuttal period, the performance of PASAT when integrated with a modern state-of-the-art CDCL solver rather than relying solely on Glucose?**
>
> Thanks for your valuable suggestion. We additionally report the performance of PASAT when integrated with CaDiCaL. Specifically, we use CaDiCaL 1.9.5, which has been adopted in recent studies such as SATFormer, and run both CaDiCaL and CaDiCaL-PASAT in plain mode. These results show that PASAT can improve the performance of CaDiCaL across the three SATCOMP benchmarks. We will include these results and provide more discussion in the revised paper.
>
> | Dataset | Solver | #Solved | #SAT | #UNSAT |
> |--------------|--------------------|--------:|-----:|-------:|
> | SATCOMP 2023 | CaDiCaL    	 |236|116|120|
> |              | CaDiCaL-PASAT |238|117|121|
> | SATCOMP 2024 | CaDiCaL    	 |252|122|130|
> |              | CaDiCaL-PASAT |256|126|130|
> | SATCOMP 2025 | CaDiCaL    	 |240|128|112|
> |              | CaDiCaL-PASAT |246|132|114|
>
>
>
> > **W2 & Q2: Can you provide an end-to-end comparison against a solver integrated with a prior neural guidance method, such as Glucose-NeuroCore?**
>
> Thanks for your valuable suggestion. We additionally report the results of Glucose-NeuroCore below. As shown in the table, Glucose-PASAT achieved performance comparable to that of Glucose-NeuroCore on SATCOMP 2023, and outperformed  Glucose-Default. Moreover, Glucose-PASAT solved 3 and 5 more instances on SATCOMP 2024/2025, respectively.
>
>
> | Dataset| Solver| #Solved | #SAT | #UNSAT |
> |--------------|--------------------|--------:|-----:|-------:|
> | SATCOMP 2023 | Glucose-Default	|     133 |   59 |     74 |
> |              | Glucose-NeuroCore|     149 |   63|     86 |
> |              | Glucose-PASAT|     148 |   64 |     84 |
> |SATCOMP 2024| Glucose-Default|     111 |   50 |     61 |
> |              | Glucose-NeuroCore|     115 |   53 |     62 |
> |              | Glucose-PASAT|     118 |   55 |     63 |
> | SATCOMP 2025 | Glucose-Default	|     135 |   60 |     75 |
> |              | Glucose-NeuroCore|     136 |   59 |     77 |
> |              | Glucose-PASAT|     141 |   62 |     79 |
>
>
> > **W3: The application of hypergraphs for SAT representation is not entirely unprecedented. However, the paper successfully mitigates this by coupling the structural representation with their novel polarity-aware decomposition mechanisms.**
>
> Thanks for this constructive comment. We will further clarify this point in the revised paper.

---

> > ### Author Rebuttal · Reviewer_RKoq · 2026-04-03
> >
> > The rebuttal has thoroughly addressed my concerns, and I am maintaining my score of 4.

---

> > > ### Author Response · Authors · 2026-04-04
> > >
> > > Thank you sincerely for your time, constructive feedback, and positive recognition of our rebuttal. We are delighted to know that your concerns have been fully resolved. We will ensure that these new results and analyses are included in the revised version. We are again truly grateful for your positive evaluation and valuable insights.

---

### Official Review · Reviewer_7qnW · 2026-03-13

**Soundness:** 2
**Presentation:** 2
**Significance:** 2
**Originality:** 2
**Overall Recommendation:** 4
**Confidence:** 3

**Summary:**

This paper addresses the problem of unsatisfiable-core variable prediction for SAT formulas and introduces PASAT, a neural architecture built on clause–literal hypergraph representations. The model incorporates a polarity-aware decomposition that separates variable embeddings into polarity-invariant and polarity-equivariant components, along with a polarity-inversion consistency regularization term. The resulting unsat-core predictions are employed to initialize variable activities in the CDCL solver Glucose.

**Compliance With Llm Reviewing Policy:**

Affirmed.

**Final Justification:**

The rebuttal adequately addresses my concerns. The authors added missing baselines, clarified differences with G4SATBench, and provided additional results on CaDiCaL, improving experimental completeness. They also included runtime and memory analysis, and strengthened comparisons with prior neural-guided approaches.

Overall, the clarifications and new results resolve my concerns, and I increase my score to 4

**Key Questions For Authors:**

1. Since the datasets follow the G4SATBench setup, why are several baselines from that benchmark not included in the comparison?

2. What is the inference time of the neural model, and how significant is the overhead when integrating it with the solver?

3. Have the authors tested the approach with other CDCL solvers (e.g., Kissat, CaDiCaL, MapleLCM)?

4. How does the solver performance compare with existing neural-guided approaches under similar experimental settings?

**Limitations:**

yes

**Strengths And Weaknesses:**

**Strengths**

1. The paper addresses an important problem in learning-guided SAT solving, predicting variables likely to belong to unsat cores.

2. The polarity-aware decomposition introduces an interesting inductive bias for modeling relationships between complementary literals.

3. The architecture is generally well described.

---

**Weaknesses**

1. The experiments use datasets derived from G4SATBench[1], but several baselines reported in the G4SATBench benchmark are not included in the comparison. As a result, it is difficult to directly assess how the proposed method compares to existing models evaluated on the same datasets.

2. The reported results appear to differ from the performance levels reported in the G4SATBench[1] paper, particularly on the CA dataset. This raises concerns about the reproducibility of the results.

3. The solver experiments only evaluate integration with Glucose. Since learning-guided heuristics can interact differently with different CDCL solvers, evaluating a single solver limits the strength of the conclusions.

4. The paper does not report the inference time of the neural model or the runtime overhead introduced when integrating it with the solver. Without this information, it is difficult to evaluate the practical benefit of the approach.

5. The solver experiments do not compare against other neural-guided SAT solving approaches. Therefore, it is unclear how much improvement comes from the proposed architecture versus the general effect of neural guidance.

[1] Li et al. "G4SATBench: Benchmarking and Advancing SAT Solving with Graph Neural Networks"

---

> ### Author Rebuttal · Authors · 2026-03-31
>
> We sincerely thank you for reviewing our paper and providing valuable suggestions. See below for our response to the your constructive comments.
>
> > **W1 & Q1 : Several baselines in the G4SATBench are not included.**
>
> Thank you for your valuable comments! We selected NeuroCore and GCN because NeuroCore is built upon NeuroSAT, while GCN is a representative message-passing GNN achieving performance comparable to that of GGNN and GIN. Nevertheless, we have additionally included the results of these baselines on the medium splits of SR/CA/PS datasets. As shown below, PASAT consistently outperforms these baselines. We will include these results in the revised paper.
>
> |Method|Precision (SR)|PR-AUC (SR)|ROC-AUC (SR)|Precision (CA)|PR-AUC (CA)|ROC-AUC (CA)|Precision (PS)|PR-AUC (PS)|ROC-AUC (PS)|
> |-|-:|-:|-:|-:|-:|-:|-:|-:|-:|
> |NeuroSAT|0.895|0.946|0.850|0.184|0.272|0.561|0.770|0.833|**0.863**|
> |GIN|0.849|0.883|0.678|0.163|0.200|0.525|0.716|0.773|0.768|
> |GGNN|0.891|0.942|0.837|0.158|0.194|0.521|0.738|0.796|0.831|
> |PASAT|**0.900**|**0.953**|**0.863**|**0.307**|**0.406**|**0.666**|**0.775**|**0.838**|0.861|
>
>
> > **W2: The reported results appear to differ from the performance levels reported in the G4SATBench.**
>
> Thanks for your insightful comment! The difference comes from different evaluation metrics. G4SATBench reports only classification accuracy, whereas our paper reports Precision, PR-AUC and ROC-AUC. For unsat-core prediction, only accuracy may be insufficient because the label distribution is often highly imbalanced. For example, simply predicting that all variables are not in the unsat core can yield an accuracy above 0.8 on the CA dataset. Therefore, these three metrics in our paper are more informative for model performance evaluation.
>
> To address the reproducibility concern, we trained a baseline using the G4SATBench source code on our generated CA dataset, and obtained an accuracy of 0.85, which is close to the 0.83 reported in their paper. We will clarify this more explicitly in the revised paper.
>
>
> > **W3 & Q3: Have the authors tested the approach with other CDCL solvers?**
>
> Thanks for your valuable suggestion! We additionally report the performance of PASAT integrated with CaDiCaL. Please kindly refer to the response to **W1 of reviewer RKoq** due to the limitation of space.
> These results show that PASAT can improve the performance of CaDiCaL across the three SATCOMP benchmarks. We will include these results in the revised paper.
>
>
> > **W4 & Q2: The paper does not report the inference time or the runtime overhead.**
>
> Thank you for raising this question!
>
> First, for unsat-core prediction, we report the computational cost of PASAT and NeuroCore on each split of the SR/CA/PS datasets, including the average inference time per instance (ms) and peak GPU memory usage (MB) during testing.
>
> |Split|Dataset|Avg. Inference Time (NeuroCore)|Peak VRAM (NeuroCore)|Avg. Inference Time (PASAT)|Peak VRAM (PASAT)|
> |---|---|---:|---:|---:|---:|
> |Easy|SR|2.61|161.48|2.89|189.58|
> ||CA|3.62|302.59|4.12|315.28|
> ||PS|3.30|207.15|3.91|236.24|
> ||Avg.|3.18|223.74|3.64|247.03|
> |Medium|SR|4.06|710.06|3.53|835.28|
> ||CA|4.16|1679.78|6.31|1730.22|
> ||PS|4.02|964.36|4.92|1084.76|
> ||Avg.|4.08|1118.07|4.92|1216.75|
> |Hard|SR|4.33|1706.10|5.69|1985.87|
> ||CA|3.88|4139.99|14.43|4160.48|
> ||PS|3.95|1912.54|11.54|2089.94|
> ||Avg.|4.05|2586.21|10.55|2745.43|
>
> Across all splits, although PASAT incurs higher inference time than NeuroCore, the inference overhead remains relatively low and practical.
> Moreover, the precision, PR-AUC and ROC-AUC of PASAT are higher than those of NeuroCore, especially on the CA dataset. And the peak GPU memory usage of PASAT remains close to that of NeuroCore.
>
> Second, in the neural-guided solver evaluation, PASAT is invoked only once before solving the instances, whereas integration methods such as NeuroCore periodically invoke the neural model. As a result, PASAT introduces only a marginal overhead for variable score generation compared with the time spent solving instances. On each SATCOMP yearly dataset, PASAT takes about 12 seconds per instance on average to generate the scores, whereas the solver takes about 2500 seconds per instance on average.
>
> We will clarify this in the revised paper.
>
> > **W5 & Q4: Compared with other neural-guided SAT solving approaches.**
>
> Thanks for your valuable suggestion! We additionally report the results of Glucose-NeuroCore. Please kindly refer to the response to **W2 of reviewer RKoq** due to the limitation of space. Glucose-PASAT achieved performance comparable to that of Glucose-NeuroCore on SATCOMP 2023, and outperformed  Glucose-Default. Moreover, Glucose-PASAT solved 3 and 5 more instances on SATCOMP 2024/2025, respectively.

---

> > ### Author Rebuttal · Reviewer_7qnW · 2026-04-04
> >
> > The rebuttal has thoroughly addressed my concerns, and I have raised my score to 4.

---

> > > ### Author Response · Authors · 2026-04-04
> > >
> > > Thank you sincerely for your time, constructive feedback, and positive recognition of our rebuttal. We are delighted to know that your concerns have been fully resolved. We will ensure that these new results and analyses are included in the revised version. We are again truly grateful for your positive evaluation and valuable insights.

---

### Official Review · Reviewer_GWMK · 2026-03-13

**Soundness:** 2
**Presentation:** 2
**Significance:** 3
**Originality:** 2
**Overall Recommendation:** 4
**Confidence:** 4

**Summary:**

This paper introduces PASAT, a new representation for SAT problems that models CNF formulas as a clause–literal hypergraph and augments it with a clause incidence graph to explicitly capture higher-order relations between clauses. Each variable representation is decomposed into a polarity-invariant and a polarity-equivariant component; message passing is performed on the hypergraph, and the results are then aggregated back into variable representations. On the training side, the authors introduce a polarity-inversion consistency regularization scheme: they construct a sign-flipped formula by negating all literals and apply contrastive learning to encourage the model to capture polarity-sensitive structure. PASAT outperforms NeuroCore, SATFormer, and other baselines on several synthetic benchmarks, and yields solver-level improvements on real SAT Competition instances when integrated into Glucose.

**Compliance With Llm Reviewing Policy:**

Affirmed.

**Final Justification:**

The rebuttal was clear and constructive, clarified the intended methodological (rather than theoretical) scope, and provided additional experiments and discussion that adequately addressed my key concerns within the rebuttal constraints. My overall assessment remains essentially unchanged, and I keep my original score.

**Key Questions For Authors:**

1. Compared with a bipartite GNN, constructing both a clause–literal hypergraph and a CIG introduces additional memory usage and computational overhead, especially for large-scale SAT instances. Have the authors considered the cost of introducing the hypergraph + CIG, and conducted a comparative analysis across different problem scales?
2. Regarding the constraint on the equivariant component in the decomposition-level consistency: in Equation (15), the term $‖V_{i,eq}^{(T)} + V_{i,eq}^{(T)(flip)}‖^2$ is used to enforce that the sum before and after flipping is close to 0. From the perspective of training dynamics, could this potentially conflict with structural difference signals learned via hypergraph message passing, or excessively compress the equivariant component, thereby impairing the model’s ability to exploit polarity differences for fine-grained discrimination?
3. The paper states that the Glucose integration experiments on SATCOMP follow the NeuroCore setup. Why is there no comparison with NeuroCore under the same integration pipeline? Including such a comparison could more clearly highlight PASAT’s advantages over GNN-based modeling at the solver level.

**Limitations:**

1. It would be helpful to discuss the method’s performance on large-scale, real-world datasets, such as highly structured industrial instances, in addition to the synthetic distributions currently considered.
2. The paper could be strengthened by analyzing the memory and computational overhead introduced by the hypergraph + CIG modeling, including how these costs scale as problem size increases.

**Strengths And Weaknesses:**

Strengths:
1. Proposes a structurally coherent modeling framework that combines a clause–literal hypergraph with a clause incidence graph, together with an invariant–equivariant decomposition and polarity-inversion consistency. The overall design is unified and internally consistent in how it handles structure and symmetry.
2. The experimental setup is reasonably systematic: it compares against GCN, NeuroCore, and SATFormer on SR / CA / PS, includes stepwise ablations and hyperparameter sensitivity studies, and further integrates PASAT into Glucose on SATCOMP 2023–2025 instances, demonstrating benefits that transfer from proxy tasks to the solver level.
3. Makes detailed and principled use of SAT-specific polarity symmetry. It explicitly constructs invariant/equivariant components and sign-flipped views, and enforces this symmetry via a decomposition-level consistency loss, which is theoretically more targeted than simply connecting complementary literals in the graph.

Weaknesses:
1. The theoretical contribution is largely empirical: although the combination of hypergraph modeling, invariant/equivariant decomposition, and consistency regularization is plausible, the paper offers little formal or quantitative analysis of why each component should yield substantial gains, e.g., in terms of representational expressiveness or how symmetry constraints reduce the search space.
2. The experiments focus mainly on synthetic distributions (SR/CA/PS) and SATCOMP benchmarks; the discussion of generalization to truly industrial-scale applications is limited, and there is no systematic study of how performance varies under different formula distributions.
3. The connection between the desired polarity-inversion consistency property and the specific form of the decomposition loss is not clearly articulated. Making this correspondence explicit would significantly improve clarity.

---

> ### Author Rebuttal · Authors · 2026-03-31
>
> We sincerely thank you for reviewing our paper and for providing valuable suggestions. Below we provide our response to the your insightful comments.
>
> > **W1: The theoretical contribution is largely empirical.**
>
> Thank you for your valuable comments. We agree that the current manuscript provides limited formal and quantitative analysis of our method. Due to the NP-completeness of SAT, neural SAT solving methods are often developed from a heuristic and empirical perspective rather than with strict theoretical guarantees. Therefore, our intention is not to present the method as a theory-driven contribution with rigorous guarantees, but rather as a SAT-specific methodological design supported by consistent empirical evidence. Our experimental results also demonstrate the effectiveness of the proposed method. We will further clarify this point in the revised paper.
>
>
> > **W2 & L1: The discussion of generalization to truly industrial-scale applications is limited. And there is no systematic study of how performance varies under different formula distributions.**
>
> Thanks for your valuable suggestion. To address the concern about industrial-scale generalization, we additionally report the unsat-core prediction performance on the SATCOMP datasets, which include structured instances from real-world domains. Due to the limited rebuttal time, we report results of SATCOMP 2023 and 2024, which include relatively smaller instances.
>
> | | | SATCOMP 2023 | | | SATCOMP 2024 | |
> |---|:---:|:---:|:---:|:---:|:---:|:---:|
> | |Precision | PR-AUC | ROC-AUC | Precision | PR-AUC | ROC-AUC |
> |NeuroCore| 0.6987 | 0.7047 | 0.7787 | 0.7239 | 0.7427 | 0.7836 |
> |PASAT| **0.7049** | **0.7123** | **0.8025** | **0.7540** | **0.7608** | **0.8533** |
>
> These results show that PASAT achieves better performance than NeuroCore on both SATCOMP 2023/2024 across all three metrics. While this does not fully cover all industrial-scale settings, it strengthens the discussion of generalization to more realistic instances.
>
> In addition, regarding different formula distributions, SR, CA, and PS are commonly used synthetic SAT datasets, and each of them follows its own generation rules and distribution [R1-R3]. Therefore, experiments on these datasets can still reflect model performance under different formula distributions, and PASAT generally outperforms the baseline methods.
>
> We will include all the above results and provide a more detailed discussion in the revised paper.
>
> [R1] Learning a SAT solver from single-bit supervision.In ICLR, 2019. \
> [R2] A modularity-based random SAT instances generator. In IJCAI, 2015. \
> [R3] Locality in random SAT instances. In IJCAI, 2017.
>
>
> > **W3 & Q2: The connection between the desired polarity-inversion consistency property and the specific form of the decomposition loss is not clearly articulated.**
>
> Thanks for your valuable comments.
> The decomposition loss is designed to directly encode the desired behavior under polarity inversion. Specifically, it encourages
>
> 1. $V_{i,inv}^{(T)} - V_{i,inv}^{(T)(flip)} \to 0$, so that the invariant component remains unchanged after flipping.
> 2. $V_{i,eq}^{(T)} + V_{i,eq}^{(T)(flip)} \to 0$, so that the equivariant component changes sign accordingly.
>
> Therefore, the loss directly matches the intended transformation properties of the two components under polarity inversion.
>
> We agree that an overly large regularization weight may over-constrain the equivariant component. To further explore the effect of this regularization term, we track the discrepancy $1-\mathrm{cos\_sim}(\cdot)$ between $V_{i,eq}^{(T)}$ and $V_{i,eq}^{(T)(flip)}$ during training. On SR, this value is around 1.6--1.7 near the end of training, and similar trends are also observed on CA and PS. This indicates that the equivariant component shows the expected opposite-direction behavior rather than collapsing. We will further clarify this point in the revised paper.
>
>
>
> > **Q1 & L2: Have the authors considered the cost of introducing the hypergraph + CIG, and conducted a comparative analysis across different problem scales?**
>
> Please kindly refer to the response to **W4 of reviewer 7qnW** due to the limitation of space.
>
>
> > **Q3: Why is there no comparison with NeuroCore under the same integration pipeline?**
>
> Please kindly refer to the response to **W2 of reviewer RKoq** due to the limitation of space.

---

> > ### Author Rebuttal · Reviewer_GWMK · 2026-04-04
> >
> > I thank the authors for their detailed and thoughtful rebuttal. Regarding W2 & L1, although I still believe that the current experimental evidence is not yet sufficient to support a fully solid and comprehensive claim about generalization to industrial-scale scenarios, I acknowledge the constraints of the rebuttal period and I appreciate that the authors have made a strong effort to supplement the experiments and discussion, including additional SATCOMP results and analysis across different formula distributions. Overall, I find the responses satisfactory and my main concerns have been addressed. I will keep my original score unchanged.

---

> > > ### Author Response · Authors · 2026-04-04
> > >
> > > Thank you sincerely for your time, constructive feedback, and positive recognition of our rebuttal. We are delighted to know that your concerns have been fully resolved. We will ensure that these new results and analyses are included in the revised version. We are again truly grateful for your positive evaluation and valuable insights.

---

### Decision · Program_Chairs · 2026-04-30

**Decision:**

Accept (regular)

**Comment:**

The paper learns a core-prediction model for SAT solver guidance: CNF formulas are modeled by a clause-literal hypergraph, augmented with representations to capture the constrainedness of literals by their inverse polarities, etc.

The experiments show the efficacy of their approach, and the authors compares against a reasonable set of baselines in a systematic way.

Some weaknesses were noted by reviewers: End-to-end evaluation is restricted to the Glucose solver and lacks other more SOTA approaches. Some vague/unsupported comments were also pointed out by reviewers; however, authors did address these in the rebuttal.